# Research on the Simplified Method of Nonlinear Finite Element Analysis for CFS-SPR Connections

**Ailin Zhang** [1]**, Zhiqiang Xie** [2,*]**, Leilei Shi** [1]**, Yanxia Zhang** [1]**, Daxing Zhou** [3] **and Xiangdong Zhang** [3]

[1] School of Civil and Transportation Engineering, Beijing University of Civil Engineering and Architecture, Beijing 100044, China
[2] School of Civil and Transportation Engineering, Beijing Energy Conservation & Sustainable Urban and Rural Development Provincial and Ministry Co-Construction Collaboration Innovation Center, Beijing University of Civil Engineering and Architecture, Beijing 100044, China
[3] China Railway Construction Group Co., Ltd., Beijing 100043, China
[*] Correspondence: xiezhiqiang@bucea.edu.cn

**Abstract:** This study reviewed some simplified methods of finite element analysis (FEA) for connections in cold-formed steel (CFS) structure, and summarized eight simplified methods divided into three categories. Shear performance tests were performed for six groups of self-piercing riveted (SPR) connection in CFS. A constitutive model of shear behavior for SPR connections was proposed, which was simplified from the load–displacement curve of shear performance test results. The models of SPR connection were established in ABAQUS by the eight simplified methods, and then the FEA results and the test results were compared. The applicable scope of each simplified model was explored, and a simplified method of FEA that was most suitable for the shear behavior of the CFS-SPR connection was proposed. Moreover, the shear performance test of the CFS shear wall with SPR was conducted by considering the rivet spacing, and failure modes and load–deformation curves were obtained. On this basis, numerical models of the CFS-SPR connection shear wall were established. By comparing the test results and the FEA results for the CFS-SPR connection shear wall, the feasibility of a simplified method of FEA applied to the CFS-SPR connection was verified. The main failure modes of the CFS-SPR connection were that the rivet tail pulled out from the bottom sheet and the rivet head pulled over from the top sheet. The SPR connection of the CFS frame could be simplified with a pin or a fastener element, and the SPR connection between the steel frame and the sheathing could be simulated by a Cartesian connector or a Spring2 element. The FEA results were highly similar to the test results for the CFS-SPR shear wall.

**Keywords:** simplified methods of FEA; constitutive model; CFS-SPR connection; shear performance; feasibility of simplified method

## 1. Introduction

Cold-formed steel (CFS) structures have the characteristics of lightweight and high strength, environmental protection, and a high degree of industrialization and assembly, which have been widely used in low and multi-rise buildings [1]. The self-piercing riveted (SPR) connection is a common sheet connection technology in the field of mechanical engineering, which has the characteristics of high bearing capacity, large stiffness, and good fatigue resistance; its connection efficiency and degree of automation, especially, are extremely high [2]. The prototyping mechanism of a SPR joint is illustrated in Figure 1 [3]. Under the action of pressure, the top sheet is pierced by the SPR joint, and the bottom sheet is pierced but not penetrated. Then the rivet skirt is spread under the guidance of a suitable die. An interlock mechanism is formed with the deformed sheets. Therefore, the two connected sheets transfer force through the interlock mechanism; this provides good shear and tensile strength for the SPR connection. Therefore, the SPR has good application potential in CFS structures, while its performance directly affected the structural

performance of the component and overall structure [4]. Therefore, the effective simulation of SPR connection directly affected the accuracy of the FEA for the entire component. Currently, numerical simulation researched on the SPR connection is mainly based on the simulation of a refined finite element model to predict shear strength, the failure mechanism, and fatigue performance [5–8], but there are few studies on the simplified model of nonlinear FEA for SPR connection. Additionally, the complex structure of the SPR joint is extremely complicated to build the refined model, and the analysis is extremely nonlinear, resulting in low efficiency and excessive time consumption of the FEA for the entire component. In addition, the structural components may require hundreds or thousands of joints, making it impossible to consider each SPR in the structure by using the refined finite element model. Therefore, it is necessary to study a simplified method of nonlinear FEA to effectively simulate the mechanical properties of the CFS-SPR connections.

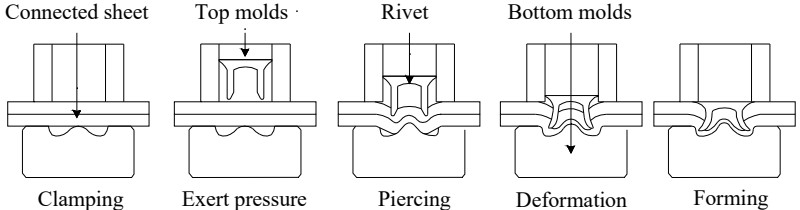

**Figure 1.** Prototyping mechanism of an SPR joint [3].

The simplified FEA methods for connections developed by scholars can be divided into three categories. The finite element models of the thin-walled sheet steel connections were developed using the commercial multipurpose nonlinear finite element program ADINA; a four-node shell element was used to model the surfaces of connecting plates and the screw, and the contacts between two surfaces were defined by three-dimensional contact elements [9]. Atsushi Kondo et al. [10] established a simplified finite element model of riveted joints by applying MSC software. The rivets were modeled with a nonlinear spring element connected to nodes located in the center of each rivet, and the relationship between shear load and relative displacement of the nonlinear springs was defined. Based on the commercial software LS-Dyna, a model considering loading rate effect was proposed by Bier and Sommer to describe the complete force–deformation behavior of the riveted connection from initial loading until failure [11]. A point-connector model based on the physical failure behavior of the SPR is developed in reference [12], which can completely simulate the load–displacement curve of the SPR connection from the beginning until the failure process.

Dourado et al. [13] proposed a simplified finite element model to represent a riveted lap joint. The rivet was built with spring-damper (combination 14) elements in ANSYS software, which could combine the stiffness constant and damping coefficient. Dubina and Fülöp [14,15] employed Combin39, a nonlinear spring element in ANSYS software that could consider slip property, to simulate the SDS connection between the CFS frame and sheathing. Furthermore, Shi et al. [16] used another simplification method, in which the SDS connection between the CFS frame and sheathing were simulated by coupling the translational degrees of freedom in the X, Y, and Z direction.

Derveni [17] employed the constraint element MPC-Pin in ABAQUS software to simulate the SDS connection between the back-to-back CFS studs and tracks, and assumed that the connection nodes are hinged. In order to achieve fixing both ends of the CFS column in ABAQUS, the constraint element MPC-Beam was adopted to simulate the rigid boundary conditions of the CFS column section [18]. Ngo [19] used the SpringA element that considers the nonlinear behavior in the ABAQUS software to simulate the SDS connection between the CFS frame and wood plate. However, the SpringA element could only consider nonlinear behavior of the axial direction. In order to simulate the simplified stiffness-equivalent model for the Magna-Lok type of rivet connection, the BUSH-type element was employed in ABAQUS to separately set the stiffness of relative

translations and rotations in individual directions [20]. Niari et al. [21] used Fastener, a mesh-independent fastener element in the ABAQUS software, to simulate the mechanical properties of the SDS connection between the CFS frame and sheathing. Attari [22] and Borzoo [23] employed the Cartesian connector element in ABAQUS software to simulate the SDS connection between the CFS frame and sheathing, which considered the nonlinear behavior of three translational directions and three rotation directions.

In general, the simplified FEA methods for connections in CFS structure could be summarized into the following three categories: (1) the shearing force was transmitted through the equivalent surface coupling in the connection area; (2) the shearing force was transmitted through the restraint element between the nodes in the connection area; and (3) the shearing force was transmitted through the link element in the joint.

The main objective of this study was to investigate a simplified FEA method, which would be suitable for the SPR connection under shear force. The application feasibility of this method was verified through numerical simulation of CFS shear walls, so as to provide a reference for the effective simulation of the SPR connection in the CFS structure. A shear performance test of SPR connection specimens was conducted to obtain failure modes and load–displacement curves. Aiming at eight types of simplified analysis methods, ABAQUS software was used to establish simplified models of SPR connection. The software was also used to perform FEA of the modeling characteristics, calculation accuracy, calculation efficiency, and failure modes of each simplified model, and compare them with the test results. On this basis, the FEA and shear performance test of the CFS shear wall with SPR were carried out. A comparative analysis was carried out between the test and the FEA results to verify the feasibility of applying the simplified finite element method to the CFS-SPR connection.

## 2. Shear Performance Test of the CFS-SPR Connection

### 2.1. Specimen Design

In this paper, taking into consideration steel sheet thickness, thickness ratio, and rivet length, six groups of CFS-SPR connection specimens were designed and manufactured, and each group of specimens was repeatedly tested with three samples. The specimens were a combination of 0.8, 1.0, 1.2, and 1.5 mm thick galvanized steel sheets. The material used for the sheets was DX51D + Z275 galvanized thin-walled steel, where "DX51D" is the number of the type of steel (acc. to EN 10142 mat. number), and "Z275" is hot galvanized with coating 275 g/m$^2$. The length of the steel sheet was 200 mm, and the width was 60 mm for the specimens. According to China specification JGJ 227-2011 [24], the end distance of the rivet was determined to be 16 mm. The test specimen of the CFS-SPR connection is shown in Figure 2.

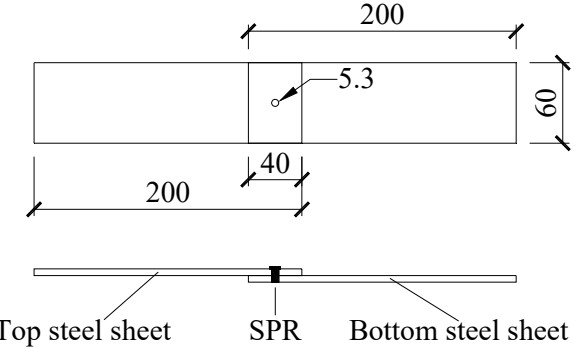

**Figure 2.** The specimens for the CFS-SPR connection (mm).

Figure 3 shows the size definition of the rivet and die used in the SPR connection, including the diameter of the rivet head ($d_1$ = 7.6 mm) and the diameter of the rivet tail ($d_2$ = 5.3 mm). The rivets were made from alloy steel with high hardness; H4 (HRC = 40–46) and H6 (HRC = 52–58) were two levels of rivet hardness measured by a

Rockwell hardness tester. In order to ensure that the rivet can pierce through the top sheet and pierce into the bottom sheet to form a reliable fastening mechanism, steel sheets of different thickness combinations need to correspond to rivets of different lengths. Based on the quality evaluation test of a large number of SPR joints in the early stage, the rivet lengths and specimens required for the combination of steel sheets with different thicknesses were determined, as shown in Table 1. The specimen numbers were defined as follows: "S0.8 + 1.5" means that the thickness of the top and bottom steel sheet was 0.8 and 1.5 mm, respectively.

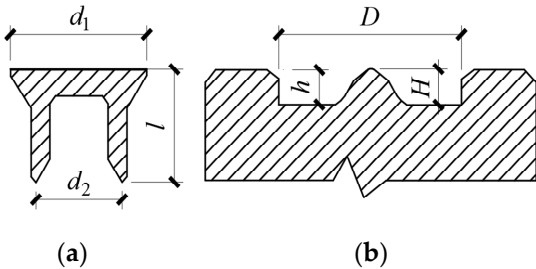

**(a)**          **(b)**

**Figure 3.** Size definition of the rivet and the bottom die: (**a**) rivet, (**b**) bottom die.

**Table 1.** Parameters of CFS-SPR connection specimens.

| Steel Sheet Combination | Length of Rivet | Inner Diameter of Die | Depth of Die | Height of Die Pip |
|---|---|---|---|---|
| | $l$/(mm) | $D$/(mm) | $h$/(mm) | $H$/(mm) |
| S0.8 + 0.8 | 4.0 | 7.0 | 1.70 | 1.70 |
| S1.0 + 1.0 | 4.5 | 7.0 | 1.70 | 1.70 |
| S1.2 + 1.2 | 5.0 | 9.0 | 1.85 | 0.55 |
| S1.5 + 1.5 | 6.0 | 9.0 | 1.85 | 0.55 |
| S0.8 + 1.5 | 4.5 | 9.0 | 1.85 | 0.55 |
| S1.0 + 1.5 | 5.0 | 9.0 | 1.85 | 0.55 |

### 2.2. Test Set-Up and Loading Protocols

The shear performance test of the CFS-SPR connection was conducted on a Zwick/ Roell Z050 testing machine equipped with an automatic extensometer, and the testing machine capacity was 50 kN (shown in Figure 4. The extensometer had a travel distance of 10 mm, which allowed the measurement of the elastic and plastic slip in the connections, and the gauge length was determined as 100 mm to measure the elongation. The monotonic loading with a loading speed of 3 mm/min was conducted [24]. The load–displacement curves were recorded and the failure modes were observed in the process of testing.

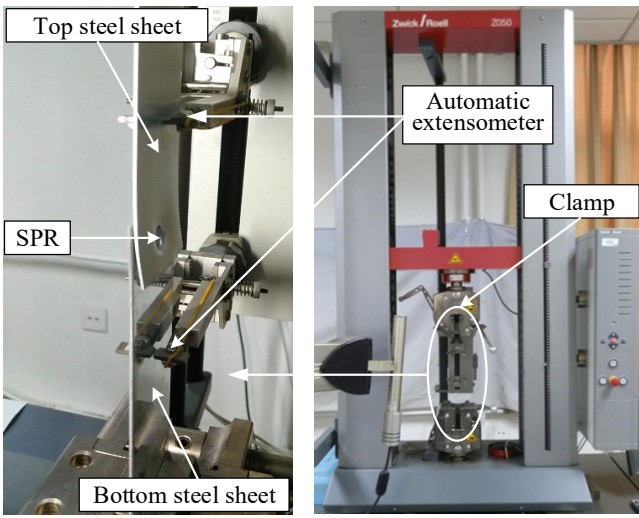

**Figure 4.** Test set-up for the CFS-SPR connection.

*2.3. Test Results*

2.3.1. Failure Modes and Mechanical Parameters

The main failure modes of CFS-SPR connection specimens under shear loading can be summarized as follows: (I) The rivet tail pulled out from the bottom sheet (see Figure 5a), and (II) the rivet head pulled over from top sheet (see Figure 5b). Table 2 presents the mechanical parameters for the six groups of CFS-SPR connection specimens.

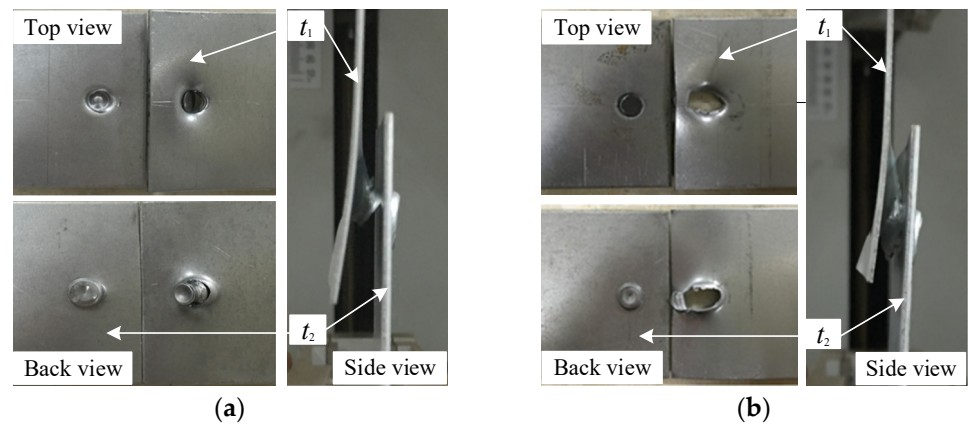

**(a)**          **(b)**

**Figure 5.** Failure modes of specimens: (**a**) failure mode I, (**b**) failure mode II.

**Table 2.** Mechanical parameters of the CFS-SPR connection.

| Specimen Number | Mean Maximum Shear Load | Mean Maximum Displacement | Failure Modes |
|:---:|:---:|:---:|:---:|
| | $P_u$/(kN) | $\Delta_u$/(mm) | |
| S0.8 + 0.8 | 3.544 | 1.023 | I |
| S1.0 + 1.0 | 4.570 | 1.063 | I |
| S1.2 + 1.2 | 6.344 | 1.121 | I |
| S1.5 + 1.5 | 7.348 | 1.151 | I |
| S0.8 + 1.5 | 4.338 | 1.180 | II |
| S1.0 + 1.5 | 5.457 | 1.722 | II |

2.3.2. Failure Mechanism of the CFS-SPR Connection

Figure 6a shows the load–displacement curve for the specimens under failure mode I. Test phenomenon indicate that the failure process of the specimens under failure mode I

can be simplified into three stages: (1) Elastic stage: the top and bottom steel sheets and the fastening mechanism of the CFS-SPR connection are in the elastic stage, and the load–displacement shows a linear growth relationship. (2) Plastic stage: the end of the top steel sheet begins to warp and deform, plastic deformation appears in the fastening mechanism. The load–displacement curve tends to be a horizontal stage. (3) Failure stage: severe plastic deformation occurs on the hole-wall of the top sheet, and the rivet is tilted and gradually pulled out from the bottom sheet. The load–displacement curve shows a sharp downward trend.

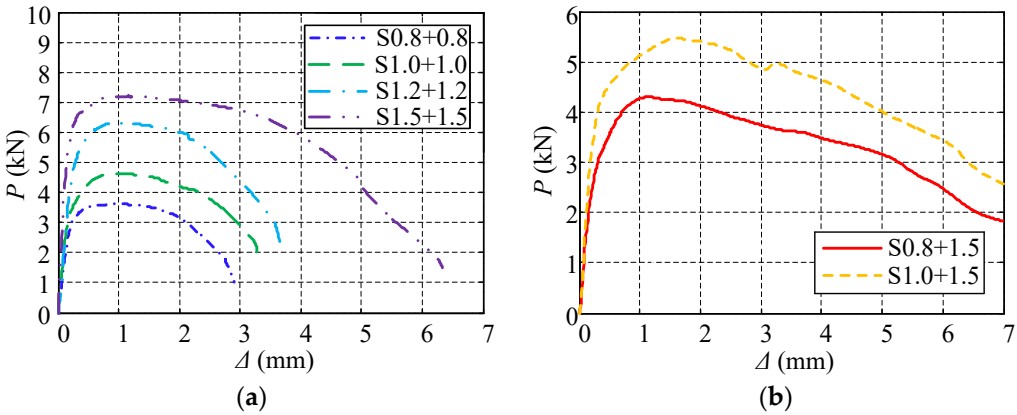

**Figure 6.** Load–relative displacement curves: (**a**) failure mode I, (**b**) failure mode II.

The failure mechanism of the CFS-SPR connection under failure mode I is as follows: the bending moment at the rivet head causes the warping deformation at the end of the top steel sheet and the tilting of the rivet, and the fastening mechanism in the bottom steel sheet shows large plastic deformation under shear loading. When the shear capacity of the fastening mechanism is less than the bearing strength of the hole-wall in the top steel sheet, the fastening mechanism fails.

Figure 6b presents the load–displacement curve for the specimens under failure mode II. In accordance with Figure 6b and the test phenomenon, the failure process of the specimens under failure mode II can be broken down into four stages: (1) Elastic stage: this stage is similar to the elastic stage of failure mode I. (2) Elastic–plastic stage: the hole-wall in the top steel sheet is locally extruded and deformed, but the fastening mechanism remains in the elastic stage. The load–displacement shows a nonlinear growth relationship. (3) Plastic stage: plastic deformation of the rivet hole increases rapidly, and the rivet head slips relative to the top steel sheet. The fastening mechanism enters elasticity–plasticity, and the load–displacement curve tends to be a horizontal stage. (4) Failure stage: the hole-wall is squeezed and torn, and the rivet head is completely pulled over from the top steel sheet. Slight plastic deformation occurred on the fastening mechanism and the load–displacement curve drops sharply.

The failure mechanism of the CFS-SPR connection under failure mode II is as follows: the fastening mechanism in the bottom steel sheet shows slight plastic deformation under shear loading. When the bearing capacity of the hole-wall is less than the shear strength of the fastening mechanism, the rivet hole is torn, and the rivet head is completely pulled over from the top steel sheet.

## 3. Simplified Methods of Nonlinear FEA Applied to the CFS-SPR Connection
### 3.1. Boundary Conditions

For the simplified analysis model, the preload generated from forming the SPR joint was ignored. The fastening force between the rivets and the friction between the two steel sheets were simplified to the shearing behavior of the connection area between the steel sheets. Figure 7 presents the boundary conditions of the simplified analysis model. All the nodes at the end of the bottom sheet release the degrees of freedom Rz rotating around the

Z axis, while all the nodes at the end of the top sheet release the degrees of freedom Rz rotating around the Z axis and the flat movement degrees of freedom Ux in the X direction (loading direction); the remaining degrees of freedom are all constrained. The nodes at the end of top steel sheet were coupled to a reference point which released the constraint in the X direction, and then a displacement was applied to the reference point in the X direction to achieve loading the model.

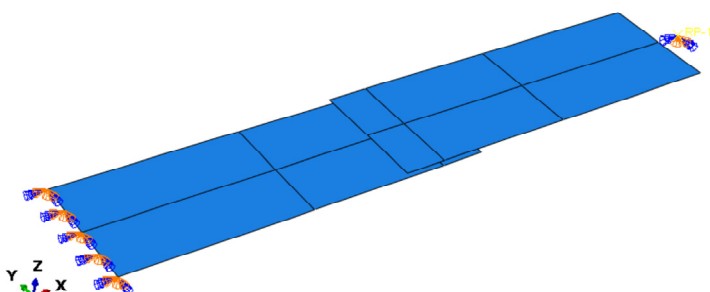

**Figure 7.** Boundary conditions of the simplified analysis model.

### 3.2. Material Properties

Based on the Chinese Standard of Metallic Materials—Tensile Testing at Ambient Temperature (GB/T228.1-2010) [25], three sets of samples from each sheet thickness were used for the tensile test, and the material properties were the average of three sets of test data. Figure 8 presents the stress–strain curves for three sets of samples for the 1.5 mm steel sheet. Tables 3 and 4 list the test results of material properties for the 1.5 and 0.8 mm steel sheets, including yield strength ($f_y$), tensile strength ($f_u$), ratio of tensile strength to yield strength ($f_u/f_y$), elastic modulus ($E$), and elongation ($S$).

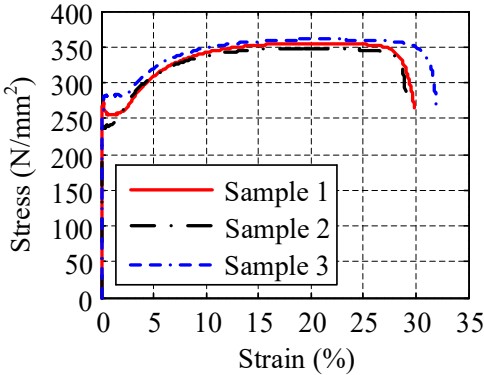

**Figure 8.** Stress–strain curves of the 1.5 mm steel sheet.

**Table 3.** Test results of material properties for the 1.5 mm steel sheets.

| Steel Sheet | $f_y$ | $f_u$ | $f_u/f_y$ | $E$ | $S$ |
|---|---|---|---|---|---|
| (t = 1.5 mm) | (MPa) | (MPa) | | (GPa) | (%) |
| Sample 1 | 255.81 | 354.84 | 1.39 | 2.06 | 30.9 |
| Sample 2 | 237.53 | 347.85 | 1.46 | 2.00 | 31.2 |
| Sample 3 | 280.64 | 361.29 | 1.29 | 2.15 | 32.1 |
| Average | 258.00 | 354.66 | 1.38 | 2.07 | 31.4 |

**Table 4.** Test results of material properties for the 0.8 mm steel sheets.

| Steel Sheet | $f_y$ | $f_u$ | $f_u/f_y$ | $E$ | $S$ |
|---|---|---|---|---|---|
| (t = 0.8 mm) | (MPa) | (MPa) | | (GPa) | (%) |
| Average | 267.68 | 361.96 | 1.35 | 2.03 | 26.6 |

The stress–strain relationship for the steel sheet obtained through the material properties test was typically the nominal stress and the nominal strain of the material. However, the material nonlinearity and geometric large deformation need to be considered in the FEA of the SPR connection. Therefore, it is necessary to convert the nominal stress and nominal strain of the material into true stress and true strain [26]. Figure 9 shows the constitutive relation between the true stress and the true plastic strain obtained by using the conversion Equations (1) and (2). On the basis of plasticity fundamentals, Equations (1) and (2) are valid only while the material is stable (before the material necking).

$$\sigma_{ture} = \sigma_{nom}(1 + \varepsilon_{nom}) \tag{1}$$

$$\varepsilon_{ture} = ln(1 + \varepsilon_{nom}) - \sigma_{ture}/E \tag{2}$$

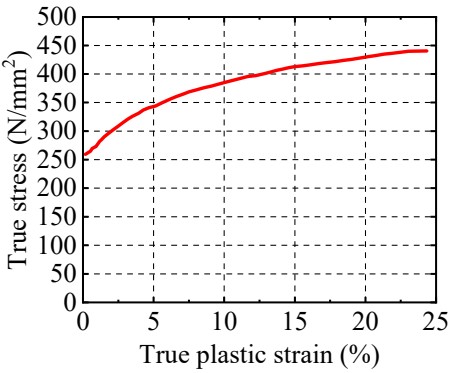

**Figure 9.** The constitutive relation between true stress and true plastic strain.

### 3.3. Shear Constitutive Model of the CFS-SPR Connection

When the SPR connection used link elements, such as the connector and the spring to simplify the simulation, it was necessary to input the constitutive relation for the connection element between the two nodes. The constitutive model of the link element was simplified according to the load–displacement curves of the shear performance test results for the CFS-SPR connection presented in Section 2. The simplified principle is presented in Figure 10a, and it demonstrates that the constitutive model of the link element can be simplified into five stages: (1) Elastic stage (OA): used to simulate the elastic behavior of the SPR connection that was to transmit force through the fastening mechanism; (2) Elastic–plastic stage (AB): used to simulate the elastic–plastic behavior caused by the warping deformation of the top steel sheet and the tilting of the rivet; (3) Plastic stage (BC): used to simulate the slip behavior between the rivet and the steel sheet; (4) Failure stage I (CD): used to simulate the behavior of the rivet separating from the sheet, causing the connection to fail completely; (5) Failure stage II (DE): used to prevent the failure of an individual rivet from causing no convergence in the calculation of the overall model, so the DE segment of the curve was treated as a horizontal segment with negligible load. Figure 10b shows the constitutive model of the shear behavior of the SPR connection in the FEA.

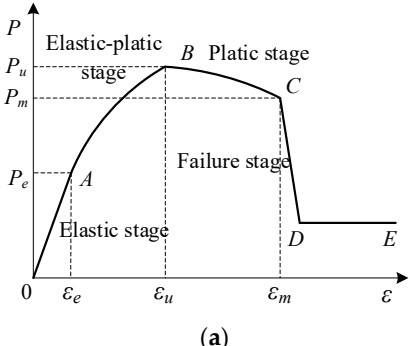
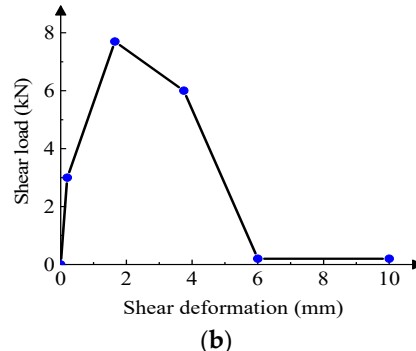

(**a**)　　　　　　　　　　　　　　　　　(**b**)

**Figure 10.** The constitutive model of the shear behavior of the SPR connection in the FEA: (**a**) simplified principle of the SPR connection, (**b**) constitutive model of the link element.

### 3.4. Simplified Methods of FEA

For the CFS-SPR connection, the simplification of the element in FEA can be divided into two types: the element simplification of the steel sheet and the element simplification of the SPR joint. The thin-shell S4R element was adopted for simplified modeling of the steel sheet. According to the different methods for transmitting shearing force, the element simplification of the SPR joint can be divided into the three categories and eight methods described in Tables 5–7.

**Table 5.** The first simplified methods of the finite element model for the SPR joint.

| Number | Simplified Sketch | Simplified Methods of Finite Element Model |
|---|---|---|
| 1-1 | Area binding Tie / Contact between sheets / Connecting sheets | The shearing behavior is equivalent to the shearing force between two rigid connecting surfaces (Area binding Tie), and assuming there is no relative movement and deformation in the binding area. The frictionless hard contact between two steel sheets is considered. |
| 1-2 | Area binding Tie / Connecting sheets | The modeling method is the same as model 1-1, but the contact between the two steel sheets is ignored. |
| 1-3 | Area coupling / Connecting sheets | The shearing behavior is equivalent to the shearing force between two coupling surfaces (Area Coupling), and the shear area between the two steel sheets is coupled to the same reference point. It is assumed that relative movement and deformation can occur in the coupling area, and the contact between the two steel sheets is ignored. The remaining parameters are the same as model 1-1. |

**Table 6.** The second simplified methods of the finite element model for the SPR joint.

| Number | Simplified Sketch | Simplified Methods of Finite Element Model |
|---|---|---|
| 2-1 | Constraint element / MPC-Pin / Connecting sheets | The rivet is simplified into a multipoint constraint element (MPC-pin), and its shear behavior is equivalent to the shearing force of the constraint element between two points [17]. The connection between the two reference points is assumed to be hinged, and the contact between the two steel sheets is ignored. |
| 2-2 | Constraint element / MPC-Beam / Connecting sheets | The rivet is simplified into a multipoint constraint element (MPC-Beam), assuming that the connection between the two reference points is rigid [18]. The remaining parameters are the same as model 2-1. |
| 2-3 | Fastener element / Fastener / Connecting plates | The rivet is simplified into a fastener element (Fastener), assuming that the connection between the two reference points is rigid [21]. The remaining parameters are the same as model 2-1. |

**Table 7.** The third simplified methods of the finite element model for the SPR joint.

| Number | Simplified Sketch | Simplified Methods of Finite Element Model |
|---|---|---|
| 3-1 | Connector element / Connecting sheets | The rivet is simplified into a Cartesian connector element, and its shearing behavior is equivalent to the shearing force of the connector element between two points [22]. The connector element can set the relative relationship of three translational and three rotational degrees of freedom between two points [23]. The contact between the two steel sheets is ignored. |
| 3-2 | Spring element / Connecting sheets | The rivet is simplified into a spring element (Spring2). The shearing area is simplified into two nodes, and its shear behavior is equivalent to the shearing force of the spring between the two points [10]. The Spring2 element can set parameters for the three orthogonal directions by setting up a local coordinate system. The remaining parameters are the same as for model 3-1. |

## 4. Simplified Analysis Results and Verification of Nonlinear FEA Model for the CFS-SPR Connection

### 4.1. Simplified Analysis Results

The thin steel sheets tended to be deformed and buckled with the increase in load; the stiffness of the SPR connection was changed, subsequently. Figure 9 presents the nonlinear relationship between true stress and true plastic strain; the frictionless hard contact between the two steel sheets was considered for simplified methods 1-1 described in Table 5; the change in contact caused the change in stiffness of the SPR connection. Therefore, the nonlinearity of materials, geometric nonlinearity, nonlinear contact of local finite element models, and Von Mises yield guidelines were considered in FEA.

For the eight simplified methods of FEA described in Tables 5–7, finite element models were established in ABAQUS, respectively. The static solution of displacement loading was adopted, and the maximum loading displacement was set as 5 mm. The main mesh size of all simplified models for the CFS-SPR connection was 3 mm. However, for the finite element models modeled by the first simplified methods, the mesh of the area around the SPR joint needed to be more intensive. The Mises stress cloud diagrams of the eight finite element models are presented in Figure 11, the failure mode of test specimens is shown in Figure 12, and the shear load–deformation curves are illustrated in Figure 13.

The analysis results for the three types of simplified finite element models shown in Figures 11–13 indicate that:

(1) The first type of finite element models that used the coupling of the equivalent surface area to simplify the simulation could accurately simulate the failure characteristics of the SPR connection. The shear load–deformation curves in the elastic stage were highly similar to the test results, but its calculations in the plastic and failure stage are difficult to converge. In addition, the load–deformation curve of model 1-1 indicated that the hard contact between the sheets had a more obvious plastic stage.

(2) The second type of three finite element models that used a constraint element to simplify modeling had a poor simulation effect on the failure characteristics of the SPR connection. There were large differences in the shear load–deformation curves of the three models, especially model 2-2 with the MPC-beam element. The shear load–deformation curve of the MPC-Pin element model was similar to the test results in the elastic, elastic–plastic and plastic stage, but that was quite different from the test results in the failure stage. For model 2-3 of the Fastener element, the shear load–deformation curve was similar to the test results in the elastic and plastic stage, but that was quite different from the test results in the elastic–plastic and failure stage.



(3) The third type of two finite element models that used a link element to simplify modeling could simulate the failure characteristics of the SPR connection very accurately. The shear load–deformation curves of the numerical simulation were in good agreement with the test results in the elastic, elastic–plastic, plastic, and failure stages.

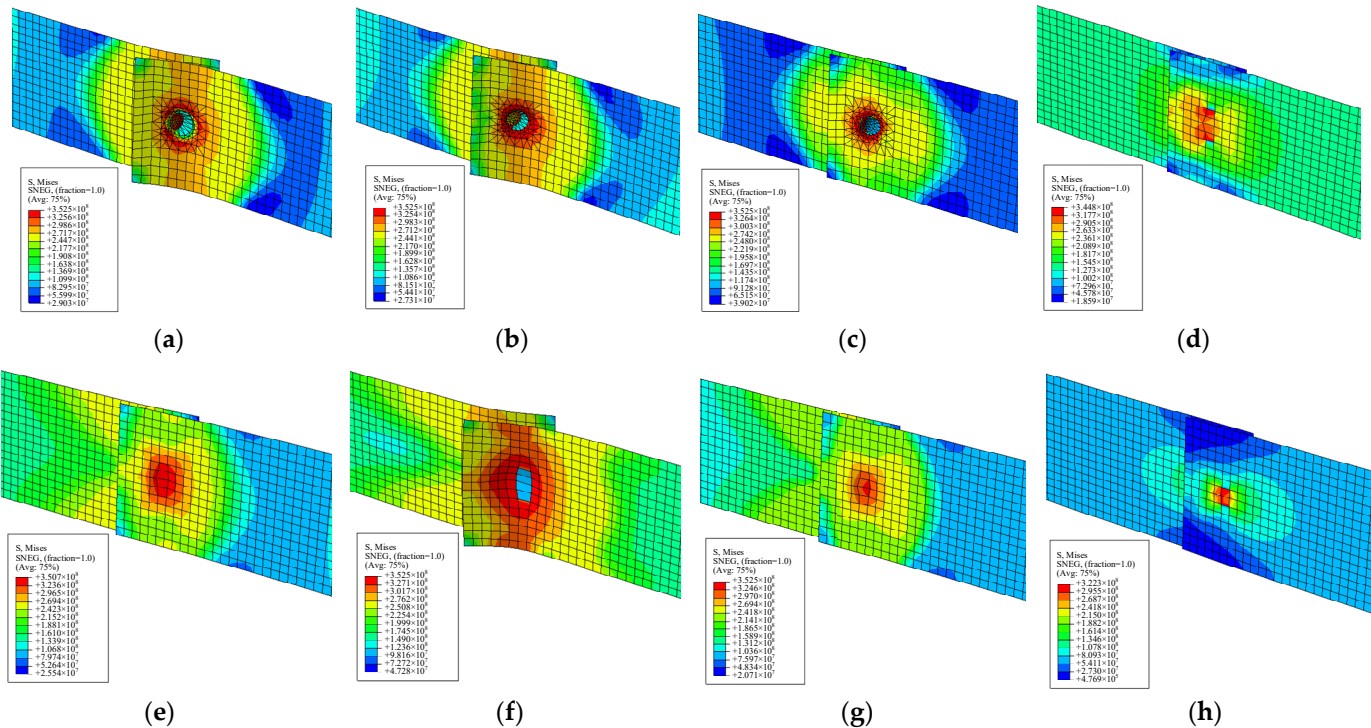

**Figure 11.** The Mises stress cloud diagrams and failure modes: (**a**) 1-1, (**b**) 1-2, (**c**) 1-3, (**d**) 2-1, (**e**) 2-2, (**f**) 2-3, (**g**) 3-1, (**h**) 3-2.

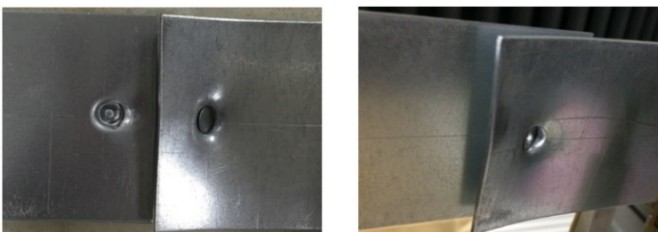

**Figure 12.** The failure mode of test specimens.

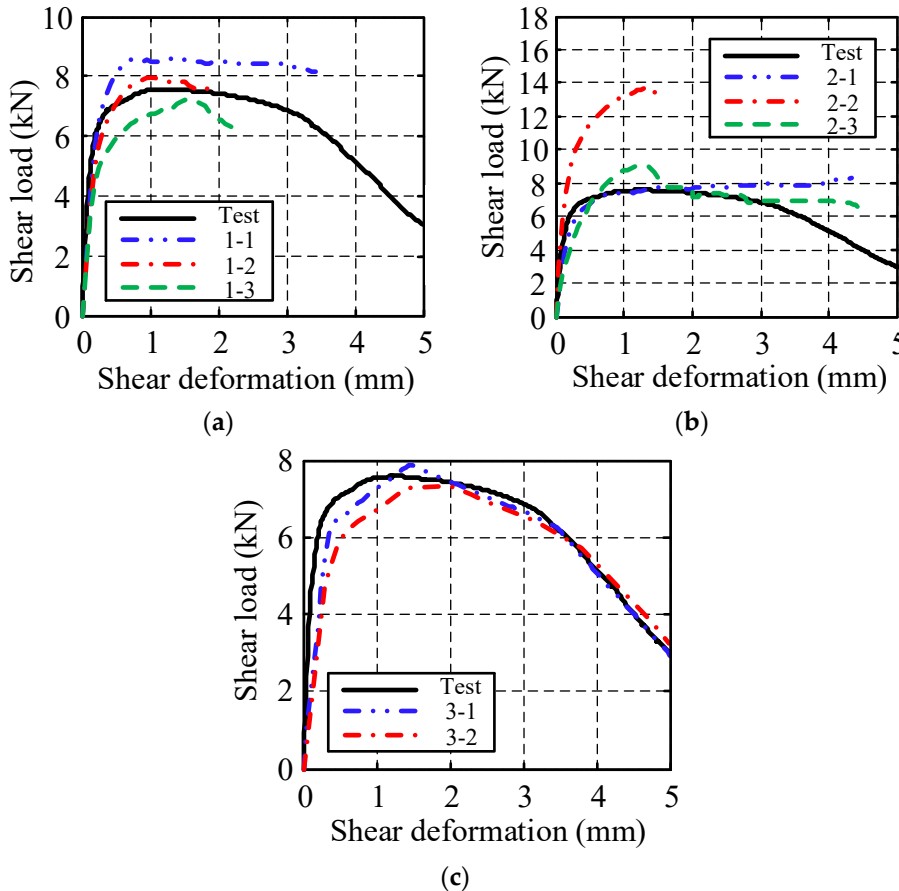

**Figure 13.** Shear load–deformation curves of the simplified model for the CFS-SPR connection: (**a**) first type of simplified model, (**b**) second type of simplified model, (**c**) third type of simplified model.

### 4.2. Verification of Analysis Results

In order to obtain the differences in the simplified analysis results, Table 8 shows a comparison of the simplified FEA models, including complexity degree of the model, similarity degree of failure modes, and calculation time. Moreover, a comparison between the FEA results and the test results is presented in Table 9.

**Table 8.** Comparison of the simplified FEA models.

| Number | Complexity Degree of Model | Complexity Degree of Mesh | Similarity Degree of Failure Modes | Calculation Time (*s*) |
|---|---|---|---|---|
| 1-1 | ★★★★ | ★★★★ | ★★★★ | 127 |
| 1-2 | ★★★ | ★★★★ | ★★★★ | 71 |
| 1-3 | ★★★ | ★★★★ | ★★★ | 58 |
| 2-1 | ★ | ★ | ★ | 23 |
| 2-2 | ★ | ★ | ★★ | 25 |
| 2-3 | ★★ | ★ | ★★★ | 43 |
| 3-1 | ★ | ★ | ★ | 10 |
| 3-2 | ★★ | ★ | ★ | 18 |

Note: The number of ★ means the degree of complexity and the degree of similarity. Computer parameters for FEA are as follows: Intel (R) Core (TM) i5-6500 CPU @ 3.20 GHz, 3201 MHz, 4 kernels, 4 logical processors, 4 GB RAM.

**Table 9.** Comparison between FEA results and test results.

| Number | $K_e/K_{test}$ | $P_u/P_{test}$ | $\varepsilon_m/\varepsilon_{test}$ | $\varepsilon_c$ (mm) |
|---|---|---|---|---|
| 1-1 | 0.79 | 1.14 | 1.08 | 3.42 |
| 1-2 | 0.61 | 1.05 | 0.80 | 2.00 |
| 1-3 | 0.46 | 0.95 | 1.17 | 2.22 |
| 2-1 | 0.90 | 1.09 | 3.38 | 4.32 |
| 2-2 | 0.92 | 1.80 | 1.02 | 1.53 |
| 2-3 | 0.43 | 1.19 | 0.98 | 4.42 |
| 3-1 | 0.86 | 1.04 | 1.14 | 5.00 |
| 3-2 | 0.74 | 0.97 | 1.58 | 5.00 |

Note: $K_e/K_{test}$ is the ratio of shear stiffness between the FEA and the test; $P_u/P_{test}$ is the ratio of the shear capacity between the FEA and the test; $\varepsilon_m/\varepsilon_{test}$ is the ratio of the maximum deformation between the FEA and the test; $\varepsilon_c$ is the ultimate calculation deformation of FEA.

The following conclusions could be obtained from the comparative analysis of Tables 7 and 8. For the finite element models that used the equivalent surface coupling to simplify modeling, the shear capacity and the plastic deformation had high calculation accuracy, and the similarity degree of failure modes was high. However, the model and the mesh were more complex and the calculation efficiency was low. Therefore, this simplified analysis method was only suitable for studying the local mechanical performance and failure characteristics of SPR joint.

For the finite element model with the MPC-Pin element, the shear stiffness and shear capacity had high calculation accuracy, and the model and the mesh were relatively simple, but the calculation accuracy of plastic deformation was low. For the finite element model with the Fastener element, the shear capacity and plastic deformation had high calculation accuracy, but the calculation accuracy of its shear stiffness was low. To summarize, the constraint elements (Pin and Fastener) were suitable for simulating the mechanical properties of the SPR connection that had not reached the failure stage. The beam element was not suitable for the nonlinear simplified FEA of the SPR connection.

For the finite element model with the link element, the shear capacity, shear stiffness, and plastic deformation had great calculation accuracy, and its model and mesh were simple. Especially, the calculation time was short. To summarize, the Cartesian connector element and the Spring2 element are suitable for studying the mechanical performance of overall members with the SPR connection. Comparing with the two methods described above, the Cartesian connector element had higher calculation efficiency and accuracy than the Spring2 element.

## 5. Feasibility Verification of the Simplified Analysis Method of SPR Connection in the CFS Shear Wall

The SPR connection in the CFS shear wall could be divided into an equal-thickness connection between the stud and track in the CFS frame and an unequal-thickness connection between the CFS frame and steel sheathing. The test results showed that the SPR connection between the stud and track in the CFS frame did not appear to be damaged, while the SPR connection between the CFS frame and steel sheathing was seriously damaged [27]. Therefore, the simulation of the SPR between the members in the CFS frame could be simplified by using the constraint element Pin or Fastener, while the SPR between the CFS frame and sheathing simulation could be simplified by using the Cartesian connector or the Spring2.

To verify the feasibility of the simplified analysis method of the SPR connection in the CFS shear wall, a shear performance test of CFS shear wall with SPR connection was conducted. Based on the simplified analysis method proposed in this paper, the finite model of the CFS shear wall was established, and FEA was performed. The FEA results were compared and analyzed with the test results.

## 5.1. Shear Performance Test of the CFS Shear Wall with SPR

The shear-wall specimens were numbered W-1 and W-2 (see Figure 14). The SPR spacing of two specimens between the CFS frame and steel sheathing was 50 and 150 mm, respectively, and the other parameters were the same. The size of the specimens was $1.2 \times 2.7$ m, the sheathing was the flat steel sheet of 0.8 mm thickness. The middle stud was C-section steel with dimensions $140 \times 40 \times 12 \times 1.5$ mm. The end stud was a combination section of two back-to-back C-section steels, and connected by double-row rivets. The track consisted of two back-to-back U-section steels with dimensions $143 \times 40 \times 1.5$ mm. The SPR spacing between the middle stud and steel sheathing was 300 mm. The shear wall was fixed to the loading bottom beam through a hold-down device. The upper end of the hold-down device was connected to the end stud by a self-drilling screw (SDS) with a diameter of 5.5 mm and a length of 55 mm, and the lower end was connected to the bottom beam by an M16 high-strength bolt.

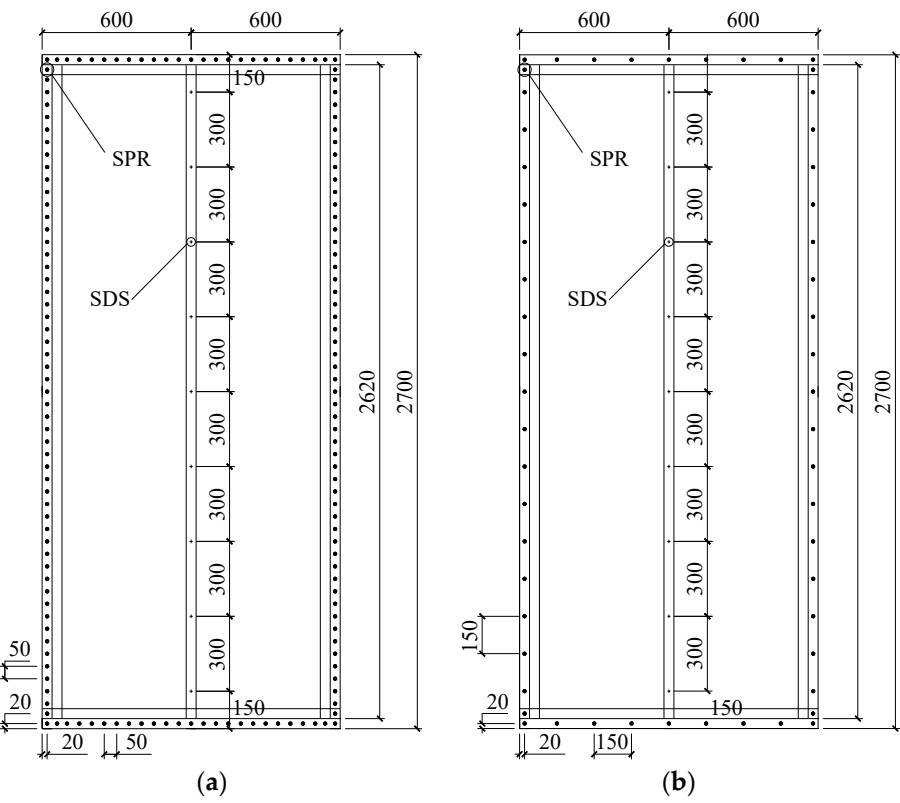

**Figure 14.** Specimens of CFS shear walls with SPR (mm): (**a**) W-1, (**b**) W-2.

## 5.2. Finite Element Models of CFS Shear Wall with SPR

For the CFS shear wall with SPR, the geometric model of each component was established, and S4R shell element was used for the CFS frame and sheathing. The position of the SPR connection was divided to obtain corresponding nodes, and the mesh of components was formed after assembly. The mesh size of the CFS frame was 10 mm and the mesh size of the steel sheathing was set to 20 mm. The established finite element models were FW-1 and FW-2, which correspond to the test specimens W-1 and W-2, respectively.

According to the comparative study of various simplified modeling methods of SPR connection in Sections 3 and 4, the Cartesian connector was selected to simulate the SPR connection in the CFS shear wall. The Cartesian connector element could simulate the relative relationship of three translational and three rotational degrees of freedom between two nodes (shown in Figure 15).

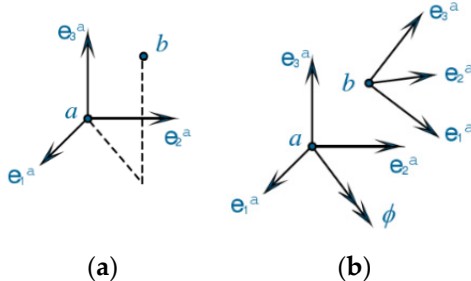

**Figure 15.** Translational and rotational relations of the Cartesian connector: (**a**) translational relation, (**b**) rotational relations.

Test results of shear walls showed the SPR connections in the CFS frame were basically intact and without any damage. In contrast, the SPR connections between the CFS frame and the steel sheathing were seriously damaged, and obvious relative slip occurred between them. Therefore, it was assumed that the SPR connection between the components in the CFS frame would be a rigid connection, and the SPR connection between the CFS frame and sheathing would be a hinged connection. The Cartesian connector element in the CFS frame was subjected to tensile load, shear load, and torsion load, so the two connected nodes needed to limit three translational and three rotational degrees of freedom. The Cartesian connector elements between the CFS frame and sheathing were only subjected to the shear load, so the two connected nodes were required to limit three translational degrees of freedom. Additionally, based on the shear constitutive model of CFS-SPR proposed in Section 3.3, the shear constitutive relationship of Cartesian connector elements was set along the shear direction of the rivet. Figure 16 shows the finite element models of the CFS shear wall with the SPR connection.

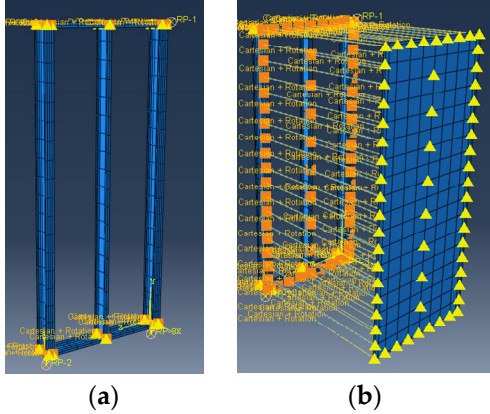

**Figure 16.** The finite element models of the CFS shear wall with SPR: (**a**) Cartesian connector elements in CFS frame, (**b**) Cartesian connector elements between the CFS frame and steel sheathing.

Figure 17 shows the simplified finite element model of the hold-down. The web in the connection area at the lower end of the end stud was set as a rigid body and coupled to a reference point. The high-strength bolt in the hold-down device was simulated by a bilinear spring (Spring2), one end of which was connected to the reference point and the other end connected to the ground.

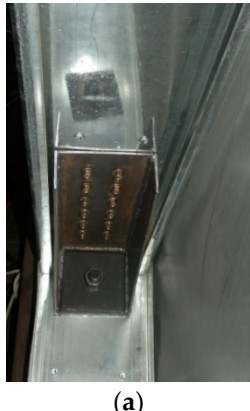
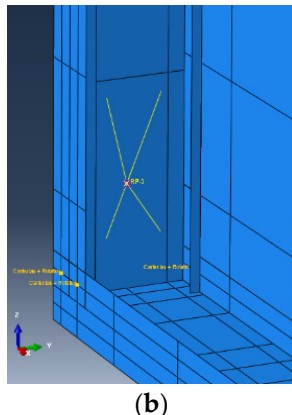

(**a**)                (**b**)

**Figure 17.** The hold-down device: (**a**) real component, (**b**) simplified finite element model.

*5.3. Comparison of Test and FEA Results*

5.3.1. Comparison of Failure Modes

Figure 18 presents a comparison of the failure modes between the FEA and the test results for specimen FW-1 (W-1) with a rivet spacing of 50 mm. It can be seen from Figure 18a that two parallel waves of shear buckling appeared along the diagonal of the sheathing, and the element stress at the wave peaks had reached the yield stress. Furthermore, some rivet heads were pulled out from the steel sheathing. The distortional buckling occurred on the bottom track at the tension side of the shear wall, and tearing between the web and flange appeared (see Figure 18b). The upper part of the end stud on the compressed side was bend-buckling, and the SPR connection at both ends of the main wave along the diagonal of the sheathing showed a large plastic deformation (see Figure 18c). Therefore, the failure modes of the shear wall with SPR were presented as shear buckling of the steel sheathing, distortional buckling of the track, bend-buckling of the end stud, and pull-out of the rivet head from the sheathing. The failure characteristics of the shear wall obtained by FEA were basically consistent with the test results.

Figure 19 shows a comparison of the failure modes between the FEA and the test results for specimen FW-2 (W-2) with the rivet spacing of 150 mm. It can be seen from Figure 18 that three parallel shear waves appeared along the diagonal of the sheathing, while the element at the wave peaks had not reached the yield stress. Moreover, the failure of the SPR occurred at the overlap between the shear wave and the edge of the CFS frame, which was especially serious at the corners of the wall. The primary failure modes of the shear wall with the rivet space of 150 mm were the shear buckling of steel sheathing and the pull-out of the rivet head from the corner of the steel sheathing. The failure models of the shear wall obtained by FEA were basically consistent with the test results. Additionally, the FEA and test results also showed that the rivet spacing at the edge of the steel sheathing had great influence on the failure modes of the shear wall.

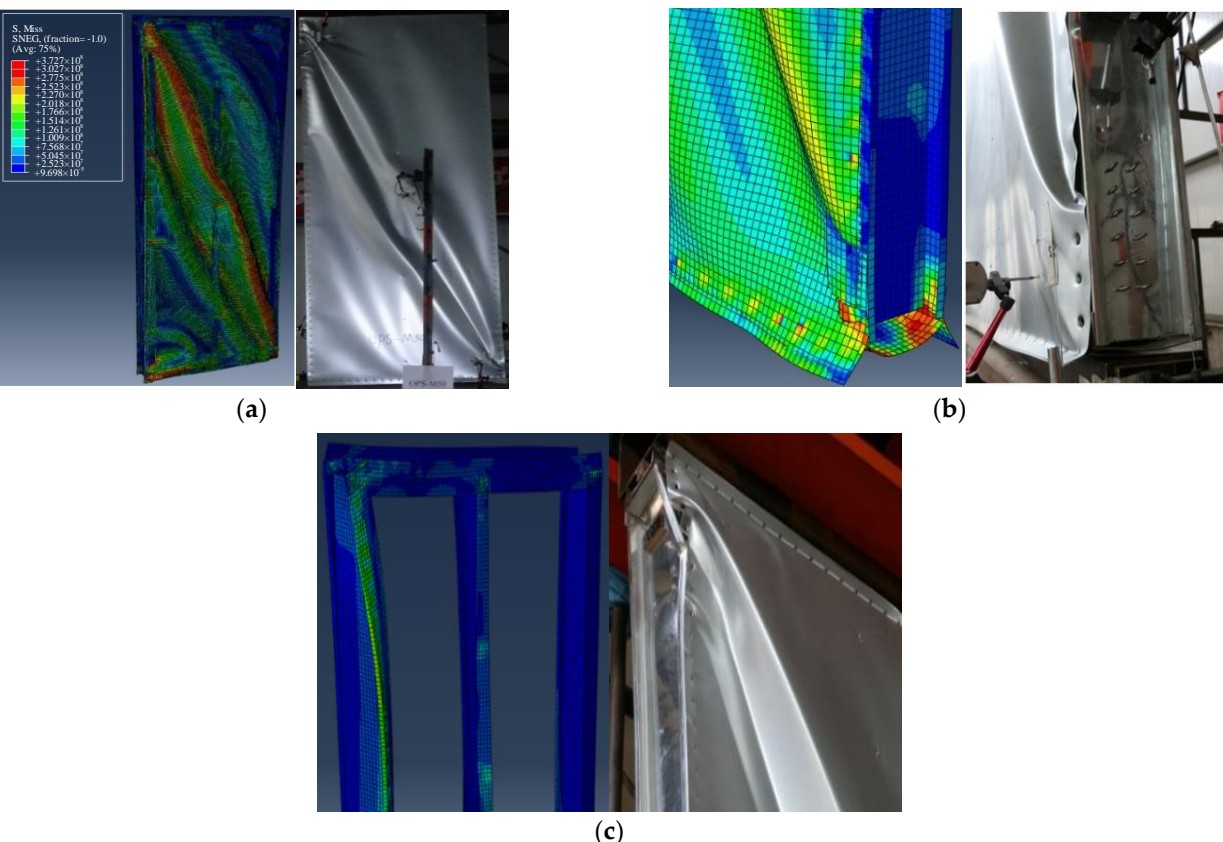

**Figure 18.** Comparison of failure modes between the FEA and the test results for specimen W-1: (**a**) shear buckling of steel sheathing and the pull-out of the rivet head from sheathing, (**b**) distortional buckling of the track, (**c**) bend-buckling of end stud.

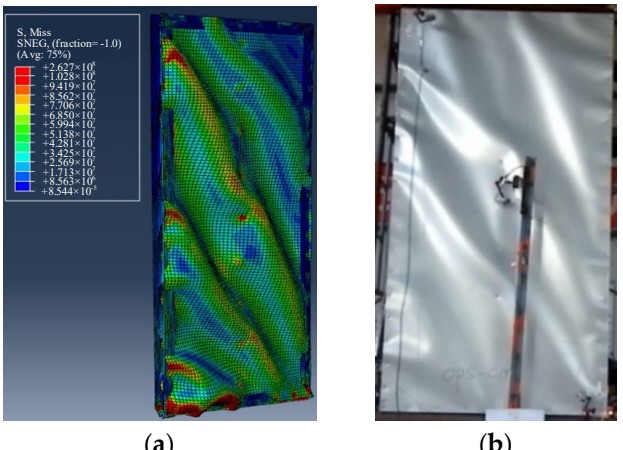

**Figure 19.** Comparison of failure modes between the FEA and the test results for specimen FW-2 (W-2): (**a**) FEA result, (**b**) test result.

### 5.3.2. Comparison of Load–Displacement Curves

A comparison of load–displacement curves between the FEA and the test is presented in Figure 20. It can be seen that the trend of the load–displacement curves for the FEA and the test was basically similar; the FEA curves agreed especially well with the test curves in the elastic stage, the elastic–plastic stage, and the plastic stage. However, the descending stage displacement of the FEA curve was smaller than that of the test curve. The main reason was that the Cartesian connector was used to simulate the SPR connection, which

the residual deformation of the hole-wall connected at SPR, and the deformation caused by the hold-down and shear bolt in the test wall could not be considered.

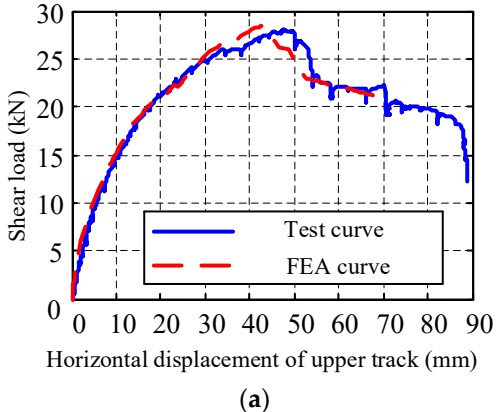

(**a**)

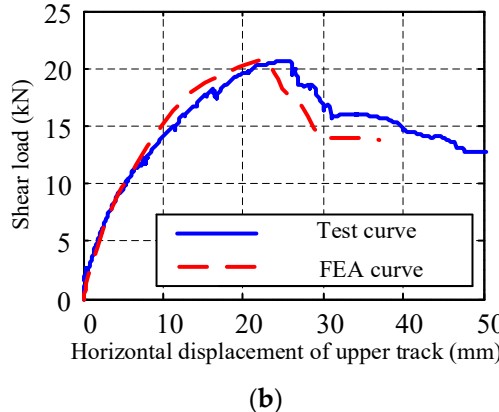

(**b**)

**Figure 20.** Comparison of load–displacement curves between the FEA and the test: (**a**) FW-1 (W-1), (**b**) FW-2 (W-2).

### 5.3.3. Comparison of Eigenvalue

A comparison of eigenvalues between the FEA and the test results is presented in Table 10. For the W-1 specimen, the yield displacement, yield load, and maximum displacement of FEA were 10.64%, 0.33%, and 12.23% less than that of the test results, respectively. The shear stiffness and shear capacity of the FEA were 12.2% and 1.39% greater than that of the test results, respectively. For the W-2 specimen, the yield displacement, yield load, and shear capacity of the FEA were 4.42%, 3.04%, and 0.53% higher than that of the test results, respectively. The maximum displacement and shear stiffness of the FEA were 13.94% and 0.53% lower than that of the test results, respectively.

**Table 10.** Comparison of eigenvalues for FEA and test results.

| Specimen Number | Results | Yield Displacement $\Delta_y$ (mm) | Yield Load $P_y$ (kN) | Maximum Displacement $\Delta_{max}$ (mm) | Maximum Load $P_{max}$ (kN) | Shear Stiffness $K_e$ (kN/m) | Shear Capacity $P_s$ (kN/m) |
|---|---|---|---|---|---|---|---|
| | FEA | 13.10 | 24.03 | 42.49 | 28.37 | 1.84 | 23.64 |
| W-1 | Test results | 14.66 | 24.11 | 48.41 | 27.98 | 1.64 | 23.32 |
| | Error (%) | 10.64 | 0.33 | 12.23 | 1.39 | 12.20 | 1.39 |
| | FEA | 8.54 | 17.96 | 22.54 | 20.74 | 1.82 | 17.28 |
| W-2 | Test results | 7.95 | 17.43 | 26.19 | 20.63 | 1.85 | 17.19 |
| | Error (%) | 7.42 | 3.04 | 13.94 | 0.53 | 1.62 | 0.53 |

In summary, the comparative analysis showed that the failure modes of the FEA and test results had high similarity; the load–displacement curves in the elastic, elastic–plastic, and plastic stages were in good agreement. In particular, the error of all the eigenvalue index between FEA and test results was controlled within 14%. Therefore, the finite element model of the shear wall based on the nonlinear simplified analysis method applied to the CFS-SPR connection had high reliability and was satisfied to the calculation accuracy in the field of civil engineering. Additionally, the simplified method of nonlinear FEA for the CFS-SPR connection proposed in this paper could provide an effective reference for the simulation of SPR connection in thin-walled steel structures.

## 6. Conclusions

This paper reviewed eight types of nonlinear simplified FEA methods divided into three categories. The corresponding simplified model of the CFS-SPR connection was

established, and FEA was conducted. Comparing FEA with the test results, a simplified FEA method that was suitable for SPR connection in thin-walled structures was proposed. Based on the proposed simplified FEA method, CFS shear-wall models with SPR were established, and the nonlinear FEA was performed. Compared with the test results for the shear wall, the feasibility of the simplified method of FEA for CFS-SPR connections in the application of CFS shear wall was verified. The principal conclusions can be summarized as follows:

(1)  The main failure modes of the CFS-SPR connection under shear loading were the pull-out of the rivet tail from the bottom sheet and the pull-over of the rivet head from the top sheet.

(2)  According to the different transmitting methods of shear force, the nonlinear simplified FEA methods applied to the CFS-SPR connection could be divided into three types: the shearing force was transmitted through the equivalent surface coupling, the shearing force was transmitted by the constraint element between the nodes, and the shearing force was transmitted through the link element between the nodes in the connection area.

(3)  The simplified analysis method of the equivalent surface coupling was only suitable for studying the local mechanical performance and failure characteristics of the SPR joint. The constraint element Pin and Fastener were suitable for simulating the mechanical properties of the SPR connection that had not reached the failure stage, while the constraint element Beam was not suitable for the nonlinear simplified FEA of the SPR. The link element Cartesian connector and the spring element Spring2 were suitable for simulating the mechanical performance of overall members with the SPR connection, and the Cartesian connector had higher calculation efficiency and accuracy than Spring2.

(4)  The SPR connection in the CFS frame could be simplified by the constraint element Pin or the fastener element Fastener, while the SPR connection between the CFS frame and steel sheathing could be simplified by using the link element Cartesian connector and the spring element Spring2.

(5)  Based on the nonlinear simplified analysis method of the CFS-SPR connection, the finite element model of the shear wall had high reliability and was satisfied to the calculation accuracy required in the field of civil engineering. Therefore, the simplified method of nonlinear FEA for CFS-SPR connection proposed in this paper could provide an effective reference for the simulation of the SPR connection in thin-walled steel structures.

**Author Contributions:** Project administration, A.Z.; writing—review and editing, Z.X.; writing—original draft preparation, L.S.; methodology, Y.Z.; software, D.Z.; resources, X.Z. All authors have read and agreed to the published version of the manuscript.

**Funding:** This research was funded by National Natural Science Foundation of China (Grant No. 52008018), R&D Program of Beijing Municipal Education Commission (Grant No. KM202110016012), Pyramid Talent Training Project of Beijing University of Civil Engineering and Architecture (Grant No. JDYC20220804), the financial support of Project funded by China Railway Construction Group Co., Ltd. (Grant No. LX22-21b), and ECERC.

**Institutional Review Board Statement:** Not applicable.

**Informed Consent Statement:** Not applicable.

**Data Availability Statement:** Not applicable.

**Conflicts of Interest:** The authors declare no conflict of interest.

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
