# Peer review of "Research on the Simplified Method of Nonlinear Finite Element Analysis for CFS-SPR Connections"

_buildings, doi:10.3390/buildings12111925_

Round 1

Reviewer 1 Report

This manuscript concerns on the simplified method of nonlinear finite element analysis for CFS-SPR connections. The manuscript is well written, thus it is suggested to be accepted after the following revisions:

1. Page 1, Title: It should be "element" instead of "el-ement".

  • 2. Table 1, 2 and 8: Please specify the units in (...)'

    3. Section 3:
    Add details of the finite elements used (type, mesh size).

4. Point 3.2: Please provide modulus of elasticity.

5. Page 13, Figure 12: Add units.

6. Page 14, Figure 14: The drawings are not very legible.

Author Response

We are sorry for the late submission of the reponse and the revised manuscripts, and we are sorry for the disruption to your work. Please see the attachment, thank you very much.

Author Response

(The authors gave the same response as above.)

Reviewer 3 Report

The paper presents the results of an experimental and numerical investigation with an adequate methodology, theoretical bases and well-represented results, as well as adequate conclusions.

There are some observations that the authors must attend to:

Review the wording of the abstract.

Provide more details on the resistance of the rivet.

Indicate how the numerical model takes care of the representation of the behavior of metal sheets against compressive stresses. Experimental results show that the behavior of thin metal elements is especially susceptible to buckling.

Author Response

(The authors gave the same response as above.)

Reviewer 4 Report

1-      Please explain the difference between the common extensometer and the automatic extensometer annotated in Fig. 3

2-      “Fig. 8 shows the constitutive relation between the real stress and the real plastic strain obtained by using the conversion formulas (2-1) and (2-2).” It is recommended to replace “real” with “true”. Check this point throughout the whole manuscript.

3-      The Eq numbering is not standard for the research articles. Is this numbering method considered based on the journal template or by the authors?

4-      Based on the plasticity fundamentals, Eq. (2-1) and (2-2) are valid only before the material instability (before necking). Hence Fig. 8 should be revised. It is recommended to trim the curve upto the maximum stress.

5-      Please discuss the required mesh density to obtain a converged result during FE. Which parameter was considered for the mesh dependency analysis?

6-      Fig. 10a,b and c show remeshing. If remeshing was defined, please give more details. Otherwise, discuss the mesh refinement close to the rivet.

7-      Since the calculation time has been reported in Table 6, it is suggested to report the utilized PC configurations such as CPU, RAM, ...

8-      It is strongly suggested to prepare a table in section 3.2 and report the utilized mechanical properties for FE simulations, including elastic and plastic properties. It is obvious that all properties should have an error bar based on the results reported in Fig. 7

Author Response

(The authors gave the same response as above.)

Round 2

Reviewer 4 Report

Thanks for addressing all comments.